# Have you heard of Rift Valley fever? Findings from a multi-country study in East and Central Africa

Raymond Odinoh[1]*, Jeanette Dawa[1,2]*, Silvia Situma[1,3], Luke Nyakarahuka[4,5,6], Luciana Lepore[7], Veerle Vanlerberghe[7], Carolyne Nasimiyu[1‡], Sheila Makiala[8‡], Christian Ifufa[8‡], Daniel Mukadi[8‡], Herve Viala[8‡], Nicholas Owor[4‡], Barnabas Bakamutumaho[4‡], Deo Ndumu[9‡], Justin Masumu[8‡], Robert F. Breiman[6,10‡], Kariuki Njenga[1,11]

**1** Washington State University Global Health Program, Nairobi, Kenya, **2** Center for Epidemiological Modelling and Analysis, University of Nairobi, Nairobi, Kenya, **3** Department of Animal Science, Pwani University, Kilifi, Kenya, **4** Uganda Virus Research Institute, Entebbe, Uganda, **5** Department of Biosecurity, Ecosystems, and Veterinary Public Health, College of Veterinary Medicine, Animal Resources, and Biosecurity, Makerere University, Kampala, Uganda, **6** Rollins School of Public Health, Emory University, Atlanta, Georgia, United States of America, **7** Institute of Tropical Medicine, Antwerp, Belgium, **8** Institut National de la Recherche Biomédicale, Kinshasa, Democratic Republic of Congo, **9** Ministry of Agriculture, Animal Industry and Fisheries, Entebbe, Uganda, **10** Infectious Diseases and Oncology Research Institute, University of Witwatersrand, Johannesburg, South Africa, **11** Paul G Allen School of Global Health, Washington State University, Pullman, Washington, United States of America

๏ These authors contributed equally to the work
‡ These authors also contributed equally to the work
* raymond.odinoh@wsu.edu (RO); jdawa@cartafrica.org (JD)

## Abstract

### Introduction

Rift Valley Fever (RVF) has caused several outbreaks across Africa, impacting human health and animal trade. Recent reports indicate sporadic detections of RVF virus among humans and animals in East Africa during inter-epidemic periods. We assessed RVF knowledge levels in East and Central Africa across countries with different epidemiological profiles.

### Materials and Methods

Individuals aged ≥10 years with acute febrile illness were enrolled from six health facilities in Kenya, Uganda, and the Democratic Republic of Congo (DRC). Socio-demographic information was collected, and participants were asked questions regarding their knowledge of RVF transmission, symptoms, prevention, and control. Blood samples were tested for anti-RVF antibodies (IgG and IgM). Knowledge was categorized as absent, basic, or advanced. Descriptive and ordinal logistic regression analysis identified factors associated with RVF knowledge.

**Data availability statement:** All relevant data underlying the findings of this study are provided in the Supporting Information files. These include de-identified survey responses, laboratory test results, and statistical outputs necessary to replicate the reported analyses. Due to ethical restrictions regarding participant confidentiality, individual-level survey data have been de-identified. Further inquiries may be directed to the PI. The data has been further uploaded to Havard repository under WSU-GHK account (https://dataverse.harvard.edu/dataverse/wsughk).

**Funding:** We acknowledge the financial support provided by the U.S. National Institute of Allergy and Infectious Diseases/National Institutes of Health (NIAID/NIH), under grant number U01AI151799, through the Centre for Research in Emerging Infectious Diseases-East and Central Africa (CREID-ECA). There was no additional external funding received for this study. The funders had no role in study design, data collection and analysis, decision to publish, or preparation of the manuscript.

**Competing interests:** The authors have declared that no competing interests exist.

## Results

Among 4,806 participants (median age 31, IQR 22–44, 57.5% female), only 20.5% had knowledge of RVF (16.4% basic, 4.1% advanced). Knowledge levels varied by country: DRC (3.1%), Uganda (16.1%), and Kenya (42.6%). RVF seropositivity was 10.4% in Uganda, with much lower rates in Kenya (2.0%) and DRC (1.5%). Factors associated with RVF knowledge included age 21–40 years (aOR 2.03; 95%CI 1,55–2.67) and >40 years (aOR 2.51; 95%CI 1.88–3.37), male gender (aOR 1.44; 95%CI 1.20–1.73), profession as a healthcare worker (aOR 5.63; 95%CI 3.48–9.12), residence in Kenya (aOR 26.8; 95%CI 15.8–48.4) or Uganda (aOR 5.43;95%CI 3.19–9.79), completing primary education (aOR 3.89; 95%CI 2.18–7.52) with advanced (postgraduate) education shown to increase knowledge, (aOR 22.8; 95%CI 4.95–18.6). Other factors included presence of livestock within the homes (aOR 1.26; 95%CI 1.01–1.57) and use of methods to prevent mosquito bites (aOR 1.62; 95%CI 1.32–1.98). Animal farmers, butchers, and those with close animal contact showed no association, despite being at-risk populations.

## Conclusion

Overall RVF knowledge was low across the study sites, with the highest levels observed in Kenya, moderate levels in Uganda despite greater exposure, and markedly low levels in the DRC. Targeted risk communication is urgently needed for high-risk populations in all regions particularly in Uganda, where elevated exposure contrasts with limited knowledge. Increased awareness is crucial for high-exposure groups in all regions, particularly in Uganda where exposure is higher, but knowledge remains relatively low.

## 1. Introduction

Rift Valley Fever (RVF) is a zoonotic viral disease that poses significant threats to human and animal health across Africa and the Middle East [1] with potential for substantial economic losses [2]. RVF virus is transmitted to humans primarily through contact with infected animal tissues or fluids and occasionally via bites from infected *Aedes* and *Culex* mosquitoes [3]. Certain factors, such as direct exposure to livestock and handling animal products, have been associated with an increased risk of severe RVF infection [4].

In East and Central Africa, RVF has been a recurring public health concern, with outbreaks reported in over 30 African countries including large outbreaks in Kenya [5] and the detection of sporadic cases in Uganda [6], [7]. Although there have been no documented reports of outbreaks in DRC [8], there is reported circulation of the virus among ruminant animals [9] and *Aedes* mosquitos [10].

The 2006–2007 RVF outbreak in Kenya resulted in 700 human cases and 170 deaths [2] with an estimated loss of USD 32 million as a result of livestock deaths,

trade bans, and reduced agricultural productivity [11]. The potential impact of large RVF outbreaks calls for effective disease surveillance, prevention, and control measures. However, the success of these measures is significantly dependant on the level of awareness and knowledge about RVF among community members.

Knowledge studies serve as crucial tools for assessing public awareness and behaviours related to specific health issues [12]. Increasing awareness can contribute to the early detection of an outbreak and help reduce the number of infected individuals and lower the peak of an epidemic [13]. While knowledge attitude and practice (KAP) studies on RVF have been conducted previously, there remains need to explore findings across different geographical regions with varying levels of disease exposure given the changing epidemiology of RVF in the recent past [14]. Our study aims to fill this gap by providing comparative multi-country insights into RVF knowledge across Kenya, Uganda, and DRC. Each country has a distinct epidemiological background regarding RVF: Kenya and Uganda have documented human outbreaks, while the DRC has not reported related haemorrhagic fever in humans to date. By examining sociodemographic characteristics, occupational factors, and differences across countries, we not only identify gaps in knowledge but also explore how these factors relate with key preventive practices.

## 2 Materials and methods

### 2.1 Study setting

The knowledge assessment was carried out as part of a 2-year longitudinal hospital-based study, conducted across three countries: Kenya, Uganda and DRC. The recruitment start dates slightly varied by country; Kenya from 28th February 2022–2nd February, 2024; Uganda from 8th October 2021–14th July, 2023 and DRC from 27th October 2021–13th July, 2023. In Kenya, the study was conducted in Murang'a county at Kigetuini dispensary and Kandara sub-county hospital in the central highlands. Uganda study sites were Kabale Regional Referral Hospital, Hamurwa Health Centre IV and Rwekubo Health Centre IV located in Kabale, Rubanda and Isingiro districts respectively in the Southwestern region. Notably, Uganda experienced an RVF outbreak in Mbarara district of western Uganda between January–March 2023, [15] which was during the study period. In DRC the study took place at Hôpital Général de Référence de Virunga, Goma located in the Eastern part of the country. The geographic distribution of the study areas, as depicted in Fig 1, highlights the diverse regions included in the study. Based on the primary study objective, which was to determine the prevalence of RVF virus antibodies among patients attending the healthcare facilities we planned to enroll 707 study participants from each facility in Kenya and Uganda, and 1,600 participants from the healthcare facility in DRC.

### 2.2 Sample size

**2.2.1 Sample size calculation in DRC.** We estimated a sample size of 1,600 human participants to be recruited over the 2 years. This calculation assumed an expected RVF seroprevalence of 3%, power of 80%, precision of 2% and confidence level of 95% [16].

**2.2.2 Sample size calculation in Kenya and Uganda.** We estimated a sample size of 707 human participants to be recruited over 2 years at each site. This calculation assumed RVF seroprevalence of 8%, a power of 80%, precision of 2%, and confidence level of 95% [17].

### 2.3 Study population

We enrolled a convenience sample of patients 10 years or older who presented with acute undifferentiated fever of (≥ 37.5°C for >24 hours and ≤28 days) at each of the healthcare facilities. To ensure the study participants were recruited throughout the year, we estimated the average recruitment rate per day to attain the required sample size over 2 years. We also enrolled all individuals with (i) unexplained bleeding or (ii) severe illness of unknown infectious aetiology lasting >7 days that was unresponsive to treatment. Persons with a clearly defined clinical disease, such as, an acute upper

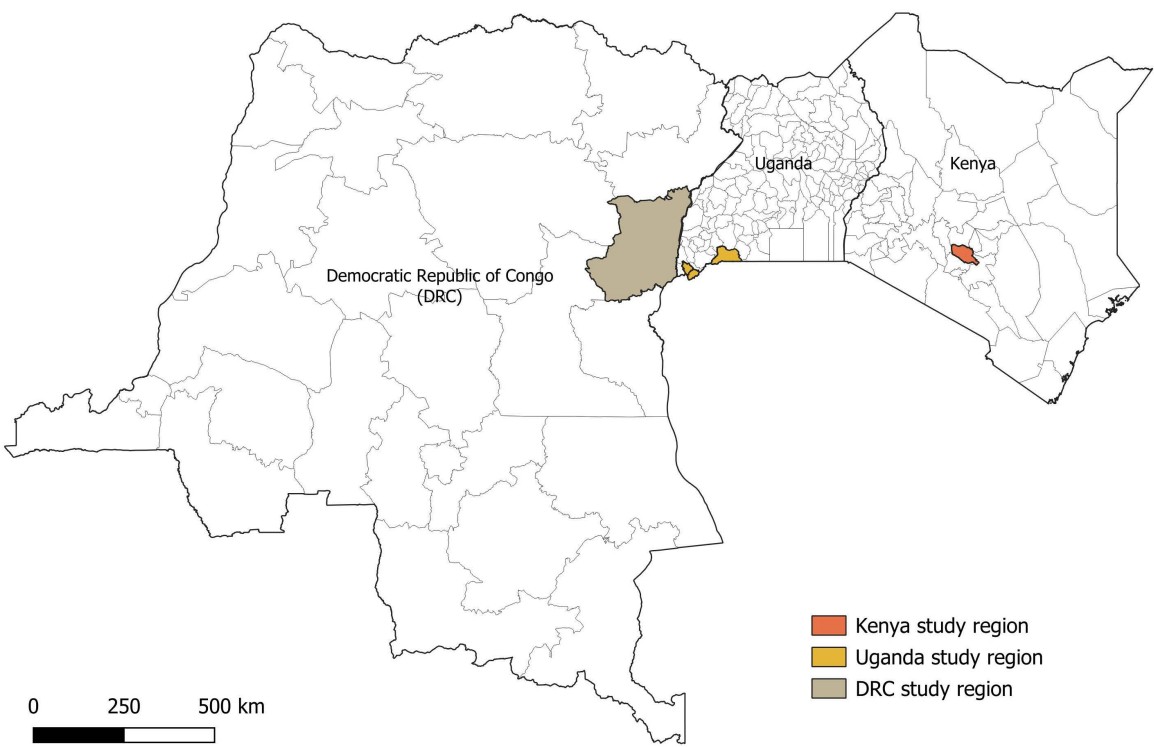

**Fig 1. Geographical map showing Rift Valley Fever study regions in Kenya, Uganda and DRC from 2021 to 2024. Shaded region (Nord Kivu, DRC): Represents the study site of Goma. Because of small geographic size of Goma, the broader region of Nord Kivu is used for better visibility. shaded regions (Kenya & Uganda): Represent study regions at comparable administrative levels.**

respiratory tract infection or urinary tract infection were not included. We enrolled patients who tested positive for malaria due to common risk factors for both malaria and RVF infection, however, we limited enrolment of patients who had positive rapid diagnostics for malaria to represent no more than 20% of the enrolled study participants.

## 2.4 Study procedures

After the initial screening, participants were taken through written consent form and allowed to ask questions before choosing to consent by signing or decline being part of the study. Both assent and parental consent forms were administered and signed for participants who were less than 18 years old and older than 10 years. Participants were handed one copy and another copy filed and stored in protected cabinets. After consenting, the study participants were guided through a standardized questionnaire designed to collect responses including socio-demographic information, risk factors related to RVF transmission and knowledge of RVF transmission, symptoms, prevention, and control (Supplementary table 1). Serum samples were collected to detect antibodies against RVF and RVF viral RNA. De-identified data was collected through the REDCap [18] by research assistants. Serum samples were initially screened for IgG and IgM antibodies against the nucleoprotein (NP) using a multi-species RVFV competitive enzyme-linked immunosorbent commercial assay (C-ELISA) from IDvet® (Grabels, France). Additionally, RVF viral RNA detection was performed on the serum samples using reverse transcription quantitative real-time PCR (RT-qPCR). All samples that tested positive in the C-ELISA were subsequently re-tested for RVF virus-specific IgM antibodies using Abbexa IgM capture ELISA (Grables, France). Details of the study procedures, including specific tests used, quality control measures, and information on sensitivity and specificity, have been previously described [19].

## 2.5 Determining knowledge scores

There were 10 knowledge related questions with either single or multiple correct answers. To assess the study participants' knowledge levels about RVF, a composite index was made by adding the scores on the individual questions. A correct response was scored as 1, while an "incorrect" or "I don't know" answer was scored as 0. In instances where there were multiple correct responses for a single question, the score would sum up to 2 depending on the number of responses the participant was able to list (Supplementary Table 1). The aggregate scores ranged from 0–14. Participants who scored an aggregate of 0 were categorized as No Knowledge, 1–10 as Basic Knowledge and 11–14 as Advanced Knowledge.

## 2.6 Data analysis

Descriptive statistics, including frequencies and percentages for each independent variable, were used to summarize data. In bivariate analysis, we used chi-square tests to assess the association between each independent variable and RVF knowledge levels. Variables with $p < 0.05$ in bivariate analysis were included in the multivariable analysis. A multivariable ordinal logistic regression model was constructed to identify independent variables associated with knowledge, while controlling for potential confounders. The backward elimination method, based on the Akaike Information Criterion (AIC), was employed for variable selection, where variables were sequentially removed from the model. This approach enabled the identification of the most parsimonious model. The proportional odds assumption of the multivariable model was assessed using the Brant test. Crude odds rations (cOR) from bivariate analysis and adjusted odds ratios (aOR) from the multivariable model along with 95% confidence intervals (CIs) were calculated and reported to measure the strength of associations. All analyses were considered statistically significant at a p-value of <0.05. Data was analysed using R statistical software version 4.4.1 [20].

## 2.7 Ethical considerations

Study ethical approval was obtained from the Kenya Medical Research Institute (KEMRI) – Research Ethics Committee (ref: SERU 4169) with a research license provided by the National Commission for Science, Technology and Innovation (NACOSTI: License No: NACOSTI/P/24/38396), Uganda Virus Research Institute (UVRI) (Ref: GC/127/849) with a research license provided by the Uganda National Council for Science and Techology (Ref: HS1713ES), Ethical Committee of the School of Public Health, University of Kinshasa, DRC, the Institut National de la Recharche Biomedicale in DRC (Ref: ESP/CE/108/2021), the Institutional Review Board of the Institute of Tropical Medicine, Antwerp and the Ethical Committee of the Antwerp University Hospital in Belgium. In addition, administrative approval was obtained from the respective Ministries of Health of each country, and from the local administration where the health facilities were located.

## Results

### Characteristics of the study population

A total of 4,806 individuals were enrolled as follows: 1,968 in Uganda, 1,468 in Kenya, and 1,370 in DRC. Of enrolled participants, 57.5%, (n = 2,763) were females; 51.9% (n = 2,494) were 21–40 years old (median age of 31 years, IQR 22–44), and 34.1% (n = 1,640) had completed primary education (Table 1). Type of occupation differed between countries, with crop farming being the most common occupation overall (34.0%), particularly prevalent in Uganda (60.8%). Animal farming was identified as the primary occupation by 13.9% of participants, although it was low in DRC (0.8%). In contrast, a larger proportion of the study participants (53.1%) reported that they had livestock (cattle, goat or sheep) kept within their homes which also varied across countries, with only 6.6% in DRC, compared to 79.1% in Kenya and 66.2% in Uganda. RVF seropositivity was 10.4% in Uganda, with much lower rates in Kenya (2.0%) and DRC (1.4%) (Table 1).

**Table 1. Characteristics of study participants in DRC, Kenya and Uganda, RVF study 2021-2024.**

| Distribution, n (%) | | | | |
|---|---|---|---|---|
| **Variable** | **Overall** | **DRC** | **Kenya** | **Uganda** |
| | **N = 4,806[1]** | **n = 1,370[1]** | **n = 1,468[1]** | **n = 1,968[1]** |
| **Age group (years)** | | | | |
| 10-20 | 925 (19.2) | 243 (17.7) | 255 (17.4) | 427 (21.7) |
| 21-40 | 2,494 (51.9) | 775 (56.6) | 679 (46.3) | 1,040 (52.9) |
| Above 40 | 1,387 (28.9) | 352 (25.7) | 534 (36.4) | 501 (25.5) |
| **Sex** | | | | |
| Female | 2,763 (57.5) | 939 (68.5) | 734 (50.0) | 1,090 (55.3) |
| Male | 2,043 (42.5) | 431 (31.5) | 734 (50.0) | 878 (44.7) |
| **Education level** | | | | |
| No education | 283 (5.9) | 101 (7.4) | 39 (2.7) | 143 (7.3) |
| Primary incomplete | 1,122 (23.3) | 128 (9.3) | 238 (16.2) | 756 (38.4) |
| Primary complete | 1,640 (34.1) | 439 (32.0) | 636 (43.3) | 565 (28.7) |
| Secondary complete | 1,194 (24.8) | 502 (36.6) | 384 (26.2) | 308 (15.7) |
| Tertiary complete | 525 (10.9) | 165 (12.0) | 169 (11.5) | 191 (9.7) |
| Postgraduate complete | 42 (0.9) | 35 (2.6) | 2 (0.1) | 5 (0.3) |
| **Occupation types** | | | | |
| Healthcare worker | 183 (3.6) | 97 (6.9) | 17 (1.0) | 69 (3.4) |
| Other professionals | 463 (9.6) | 169 (12.0) | 144 (9.9) | 150 (7.7) |
| Unskilled workers | 530 (11.1) | 111 (8.2) | 280 (19.3) | 138 (7.1) |
| Animal farmer | 669 (13.9) | 11 (0.8) | 355 (24.1) | 302 (15.3) |
| Crop farmer | 1,634 (34.0) | 37 (2.6) | 400 (27.2) | 1,196 (60.8) |
| Butcher/Slaughterhouse worker | 33 (0.6) | 3 (0.2) | 15 (0.8) | 15 (0.7) |
| Undisclosed | 26 (0.5) | 8 (0.5) | 2 (0.1) | 16 (0.8) |
| Student | 619 (12.9) | 233 (17.0) | 155 (10.6) | 230 (11.7) |
| **Presence of livestock within household** | | | | |
| No | 2,253 (46.9) | 1,280 (93.4) | 307 (20.9) | 666 (33.8) |
| Yes | 2,553 (53.1) | 90 (6.6) | 1,161 (79.1) | 1,302 (68.2) |
| **Slaughter sick livestock** | | | | |
| No | 241 (80.6) | 1 (100.0) | 94 (85.5) | 146 (77.7) |
| Yes | 58 (19.4) | 0 (0.0) | 16 (14.5) | 42 (22.3) |
| **Mosquito prevention at household** | | | | |
| No | 2,224 (46.3) | 749 (54.7) | 996 (67.8) | 479 (24.3) |
| Yes | 2,582 (53.7) | 621 (45.3) | 472 (32.2) | 1,489 (75.7) |
| **Drinking raw milk** | | | | |
| No | 4,600 (95.7) | 1,331 (97.2) | 1,448 (98.6) | 1,821 (92.5) |
| Yes | 206 (4.3) | 39 (2.8) | 20 (1.4) | 147 (7.5) |
| **Close contact with livestock** | | | | |
| No | 495 (10.3) | 251 (18.3) | 55 (3.7) | 189 (9.6) |
| Yes | 4,311 (89.7) | 1,119 (81.7) | 1,413 (96.3) | 1,779 (90.4) |
| **Close proximity to a swamp/quarry/irrigation scheme** | | | | |
| No | 2,393 (49.8) | 1,169 (85.1) | 607 (41.5) | 617 (31.3) |
| Yes | 2,413 (50.2) | 201 (14.9) | 861 (58.5) | 1,351 (68.7) |
| **RVF seropositivity (IgM/IgG)** | | | | |
| Negative | 4,553 (94.7) | 1,351 (98.6) | 1,439 (98.0) | 1,763 (89.6) |
| Positive | 253(5.3) | 19 (1.4) | 29 (2.0) | 205 (10.4) |

[1]n (%)

## Exposure factors for RVF infection

Of the known exposure factors for RVF infection, presence of livestock within the homes, was reported by 53.1% of all participants, including 79.1% Kenyans, 68.2% Ugandans, and 6.6% in DRC participants. Close contact with livestock, a known risk factor was common across all three countries (89.7% overall), this was reported even among participants who did not own livestock. Types of close contact included herding, milking, slaughtering, handling raw meat, cleaning livestock sheds, sleeping in the same room as livestock, feeding, and treating or spraying livestock. About half (50.2%) of the participants lived near a swamp, quarry or irrigation scheme, including 68.7% of Ugandans, 58.5% of Kenyans, and 14.9% of DRC participants. Of all participants, 19.4% reported involvement in slaughtering sick animals, primarily in Uganda (22.3%) and Kenya (14.5%). Only a minimal number (4.3%) of participants reported drinking raw milk. Nearly half of study participants (46.3%) did not use any methods to prevent mosquito bites at home including 67.8% in Kenya, 54.7% in DRC and 24.3% in Uganda (Table 1).

## RVF knowledge

As shown in Table 2, only 1 in 5 participants had heard of RVF, including 42.6% Kenyans, followed by 16.1% Ugandans and 3.1% DRC participants. Of the participants that had heard of RVF, 58.3% knew humans can be infected by the RVF virus. The top responses on how RVF can be transmitted in humans included eating raw meat from sick animals and mosquito bites at 37.3% and 33.2% respectively. Sixty-six point five percent knew RVF virus can infect livestock, 75.6% believing that animal infection can be prevented with vaccination (84%) listed as the top preventive method.

## Factors associated with RVF knowledge

In bivariate analysis, age was an important factor associated with knowledge on RVF, with individuals aged 21–40 years (cOR: 1.93, 95%CI: 1.56–2.41, p < 0.001) and those above 40 years (cOR: 2.27, 95%CI: 1.81–2.86, p < 0.001) more likely to have advanced knowledge compared to the 10–20 years' age group. Males had higher odds of advanced knowledge (cOR: 1.83, 95%CI: 1.59–2.11, p < 0.001). Education level showed positive association with RVF knowledge, with tertiary education completion having the highest odds ratio (cOR: 9.33, 95%CI: 5.69–16.30), (Supplementary Table 2).

Occupational categories showed varying associations with RVF knowledge levels. Healthcare workers had significantly higher odds of advanced knowledge (cOR: 4.42, 95%CI: 3.22–6.06, p < 0.001). Positive associations were also observed in other professionals (cOR: 1.45, 95%CI: 1.17–1.80, p < 0.001) and animal farmers (cOR: 1.88, 95%CI: 1.58–2.25, p < 0.001). Unskilled workers also showed higher odds of better knowledge (cOR: 1.68, 95%CI: 1.37–2.04, p < 0.001), while crop farmers and butchers/slaughterhouse workers were not significantly associated with high degrees of knowledge. Seropositive individuals showed marginally higher odds of having better knowledge (cOR: 1.12, 95% CI: 0.82–1.51, p = 0.14), (Supplementary Table 2).

Presence of livestock within homes was strongly associated with better knowledge (cOR: 3.18, 95% CI: 2.72–3.72, p < 0.001), as was close contact with animals (cOR: 2.83, 95% CI: 2.09–3.94, p < 0.001). The presence of swamps, quarries, or irrigation schemes in the vicinity of the home was also associated with higher knowledge levels (cOR: 4.82, 95% CI: 3.97–5.90, p < 0.001). Practicing mosquito prevention was also associated with better knowledge (cOR: 1.20, 95% CI: 1.05–1.39, p < 0.001), while drinking raw milk showed negative association (cOR: 0.61, 95% CI: 0.40–0.90, p = 0.021). There were also differences in knowledge levels across countries, with Kenya showing the highest odds of better knowledge compared to DRC (cOR: 21.40, 95% CI: 15.70–30.00, p < 0.001), (Supplementary Table 2).

In multivariable analysis, age was associated with advanced RVF knowledge, individuals aged 21–40 years (aOR: 2.03, 95% CI: 1.55–2.67, p < 0.001) and those above 40 years (aOR: 2.51, 95% CI: 1.88–3.37, p < 0.001) were more likely to have higher knowledge levels compared to the 10–20 years' age group. Males were more likely to have higher knowledge levels compared to females (aOR: 1.44, 95% CI: 1.20–1.73, p < 0.001) (Table 3).

**Table 2. Distribution of Rift Valley Fever knowledge and information sources among study participants in DRC, Kenya, and Uganda.**

Distribution, n (%)

| Variable | Overall | DRC | Kenya | Uganda |
|---|---|---|---|---|
| **Have you heard of RVF** | | | | |
| No | 3821 (79.5) | 1328 (96.9) | 842 (57.4) | 1651 (83.9) |
| Yes | 985 (20.5) | 42 (3.1) | 626 (42.6) | 317 (16.1) |
| **Can humans be infected with RVF virus? (n = 982)** | | | | |
| I don't know | 332 (33.9) | 3 (7.1) | 251 (40.3) | 78 (24.6) |
| No | 77 (7.8) | 0 (0.0) | 69 (11.1) | 8 (2.5) |
| Yes | 573 **(58.3)** | 39 (92.9) | 303 (48.6) | 231 (72.9) |
| **Knowledge of how RVF is transmitted in humans (multiple responses, n = 573)** | | | | |
| Mosquito bites | 190 (33.2) | 36 (92.3) | 47 (15.6) | 107 (46.3) |
| Eating raw meat from a sick animal | 214 **(37.3)** | 33 (77.0) | 111 (36.7) | 70 (30.3) |
| Drinking raw milk from a sick animal | 166 (30.0) | 30 (76.9) | 88 (29.0) | 48 (20.8) |
| Slaughtering/skinning sick animals | 126 (22.0) | 28 (71.8) | 66 (21.8) | 32 (13.9) |
| Handling abortus | 59 (10.3) | 28 (71.8) | 23 (7.6) | 8 (3.5) |
| Milking sick animals | 80 (14.0) | 21 (53.8) | 40 (13.2) | 19 (8.2) |
| Contact with blood of a sick animal | 132 (23.0) | 26 (66.7) | 63 (20.8) | 43 (18.7) |
| **Can RVF be prevented in humans? (n = 571)** | | | | |
| I don't know | 217 (38.0) | 2 (5.1) | 154 (51.0) | 61 (26.5) |
| No | 12 (2.1) | 1 (2.6) | 6 (2.0) | 5 (2.2) |
| Yes | 342 **(59.9)** | 36 (92.3) | 142 (47.0) | 164 (71.3) |
| **Knowledge of prevent RVF in humans (among respondents aware that RVF is preventable, n = 342)** | | | | |
| Avoid consuming uninspected meat/raw milk | 242 **(70.2)** | 35 (94.3) | 105 (73.9) | 102 (61.6) |
| Use protective gear for aborted materials | 84 (24.3) | 30 (80.6) | 33 (23.2) | 21 (12.8) |
| Avoid contact with fluids from sick animals | 153 (44.4) | 32 (86.1) | 74 (52.1) | 47 (28.7) |
| Drain stagnant waters/clearing bushes | 43 (12.3) | 24 (66.7) | 5 (3.5) | 14 (8.0) |
| **Can animals be infected with the RVF virus? (n = 980)** | | | | |
| I don't know | 293 (29.9) | 4 (9.5) | 195 (31.4) | 94 (29.8) |
| No | 35 (3.6) | 0 (0.0) | 22 (3.5) | 13 (4.1) |
| Yes | 652 **(66.5)** | 38 (91.5) | 405 (65.1) | 209 (66.1) |
| **Knowledge of how RVF is transmitted in animals (multiple responses, n = 650)** | | | | |
| Mosquito | 171 **(26.3)** | 34 (91.9) | 40 (10.0) | 97 (46.6) |
| Ticks | 73 (11.2) | 13 (35.1) | 0 (0.0) | 60 (28.8) |
| Biting flies | 42 (6.5) | 19 (51.4) | 3 (0.7) | 20 (9.6) |
| **Can RVF be prevented in animals? (n = 655)** | | | | |
| I don't know | 86 (13.1) | 0 (2.6) | 69 (17.6) | 12 (6.2) |
| No | 74 (11.3) | 0 (0.0) | 38 (9.3) | 36 (17.2) |
| Yes | 495 **(75.6)** | 37 (97.4) | 298 (73.0) | 160 (76.6) |
| **Knowledge of ways to prevent RVF in animals** | | | | |
| Vaccination | 416 **(84.0)** | 27 (73.0) | 282 (94.6) | 107 (66.9) |
| Treatment | 283 (57.1) | 12 (12.4) | 202 (67.8) | 69 (43.1) |
| Avoiding contact with sick herd | 54 (10.9) | 32 (86.5) | 9 (3.0) | 13 (8.1) |
| Quarantine | 94 (19.0) | 29 (78.4) | 3 (1.0) | 62 (38.8) |

[1]n (%)

**Table 3. Multivariable model results showing factors associated with RVF knowledge among study participants in DRC, Kenya and Uganda (2021-2024).**

| Variable | Crude Odds Ratios | | | Adjusted Odds Ratios | | |
|---|---|---|---|---|---|---|
| | cOR[1] | 95% CI[1] | p-value | aOR[1] | 95% CI[1] | p-value |
| **Age group (years)** | | | | | | |
| (ref 10–20) | – | – | <0.001 | – | – | <0.001* |
| 21-40 | 1.93 | 1.56, 2.41 | | 2.03 | 1.55, 2.67 | |
| Above 40 | 2.27 | 1.81, 2.86 | | 2.51 | 1.88, 3.37 | |
| **Sex** | | | | | | |
| (ref = Female) | – | – | <0.001 | – | – | <0.001* |
| Male | 1.83 | 1.59, 2.11 | | 1.44 | 1.20, 1.73 | |
| **Education Level** | | | | | | |
| (ref = No education) | – | – | <0.001 | – | – | <0.001* |
| Primary incomplete | 2.06 | 1.26, 3.59 | | 1.58 | 0.87, 3.09 | |
| Primary complete | 4.37 | 2.73, 7.51 | | 3.89 | 2.18, 7.52 | |
| Secondary | 4.78 | 2.96, 8.24 | | 8.04 | 4.41, 15.8 | |
| Tertiary | 9.33 | 5.69, 16.3 | | 9.26 | 4.95, 18.6 | |
| Postgraduate | 6.02 | 2.45, 14.3 | | 22.8 | 4.95, 18.6 | |
| **Healthcare workers** | | | | | | |
| (ref = No) | – | – | <0.001 | – | – | <0.001* |
| Yes (ref = No) | 4.42 | 3.22, 6.06 | | 5.63 | 3.48, 9.12 | |
| **Country** | | | | | | |
| (ref = DRC) | – | – | <0.001 | – | – | <0.001* |
| Kenya | 21.4 | 15.7, 30.0 | | 26.8 | 15.8, 48.4 | |
| Uganda | 6.06 | 4.41, 8.54 | | 5.43 | 3.19,9.79 | |
| **Presence of Livestock within households** | | | | | | |
| (ref = No) | – | – | <0.001 | – | – | 0.01 |
| Yes | 3.18 | 2.72, 3.72 | | 1.26 | 1.01, 1.57 | |
| **Preventing mosquito bites** | | | | | | |
| (ref = No) | – | – | 0.007 | – | – | <0.001* |
| Yes (ref = No) | 1.20 | 1.05, 1.39 | | 1.62 | 1.32, 1.98 | |
| **Drinking raw milk** | | | | | | |
| (ref = No) | – | – | 0.011 | – | – | 0.111 |
| Yes | 0.61 | 0.40, 0.90 | | 0.83 | 0.51, 1.29 | |
| **Having close contact with animals** | | | | | | |
| (ref = No) | – | – | <0.001 | – | – | 0.108 |
| Yes | 2.83 | 2.09,3.94 | | 1.42 | 0.92,2.28 | |
| **Presence of swamp/quarry/irrigation scheme** | | | | | | |
| (ref = No) | – | – | <0.001 | – | – | <0.001* |
| Yes | 4.82 | 3.97, 5.90 | | 5.80 | 4.60, 7.36 | |

Compared to those with no education, individuals with postgraduate education had the highest odds of better knowledge (aOR: 22.8, 95% CI: 4.00–32.4, p < 0.001), followed by those with tertiary (aOR: 9.26, 95% CI: 4.95–18.6, p < 0.001) and secondary education (aOR: 8.04, 95% CI: 4.41–18.60, p < 0.001). Being a healthcare worker was also associated with higher knowledge levels (aOR: 5.63, 95% CI: 3.48–9.12, p < 0.001). Country of residence was also a significant factor, with participants from Kenya (aOR: 26.8, 95% CI: 15.8–48.4, p < 0.001) and Uganda (aOR: 5.43, 95% CI: 3.19–9.79, p < 0.001) showing higher odds of knowledge compared to those from DRC (Table 3).

Environmental factors such as presence of swamps, quarries, or irrigation schemes near the subject's home were associated with higher knowledge levels (aOR: 5.80, 95% CI: 4.60–7.36, p < 0.001). Individuals who reported preventing mosquito bites were likely to have higher knowledge levels (aOR: 1.62, 95% CI: 1.32–1.98, p < 0.001). Respondents who reported presence of livestock within their homes were likely to have better knowledge levels (aOR: 1.26, 95%CI 1.01–1.57, p = 0.01). Other factors such as RVF seropositivity, drinking raw milk, other occupation categories, and having close contact with livestock did not show statistically significant associations with RVF knowledge levels in this multivariable model (Table 3).

## Discussion

In this study, we evaluated knowledge of RVF among 4,806 individuals across Kenya, Uganda, and the DRC. Our findings indicate that levels of knowledge related to RVF were low, highlighting significant gaps that may hinder effective disease prevention and control efforts in these regions despite ongoing transmission of the disease. Factors associated with better RVF knowledge included age, education level, sex, occupation, country of residence, presence of a water body near the home and those who are proactive about preventing mosquito bites.

We observed better RVF knowledge in older age groups, suggesting that cumulative exposure to information may play a key role in enhancing awareness. In Baringo, Kenya families with older household heads had greater RVF knowledge compared to families with younger household heads [13]. Similarly a study among livestock farmers in Malawi found that those over 45 years of age demonstrated better knowledge and attitudes toward the disease [21]. This is also consistent with various studies in Uganda among slaughterhouse workers and community members in Kabale district which found out that older individuals were more knowledgeable about RVF [22]. An Ebola and Marburg virus knowledge study in Uganda also showed that older age was associated with greater awareness [23], reinforcing the link between age and knowledge. These findings indicate a potential vulnerability to RVF among younger populations especially given the lack of large outbreaks in the recent past and points to the need for targeted educational interventions aimed at younger age groups.

Education showed strong association with RVF knowledge, with higher levels of education corresponding to increased awareness. Postgraduate education was associated with eleven times higher awareness of RVF highlighting the significant impact of advanced education on RVF knowledge. This finding aligns with recent research conducted in Tanzania, which demonstrated that RVF knowledge is significantly related to sex, education, and locality [24]. The disparities in RVF knowledge across different educational levels shows the importance of making RVF information accessible and understandable to those with lower education levels possibly through community-based education programs. Community-based education programs in West Africa during the Ebola outbreak were instrumental in increasing public understanding of the disease, leading to improved prevention practices and a reduction in the spread of the virus [25].

While healthcare workers exhibited higher levels of knowledge, likely due to their training, this was not the case for other occupational groups, including those at greater risk of exposure. An important finding of the study is the significant gap in RVF knowledge among high-risk occupational groups, such as animal farmers, butchers and slaughterhouse workers. This brings out the need for targeted interventions to raise RVF awareness among these higher at-risk groups, ensuring that those considered highly vulnerable are adequately informed and protected.

Participants from Kenya and Uganda demonstrated greater likelihood for higher levels of RVF knowledge compared to those from DRC. Kenya has had recurrent RVF epidemics with cases documented in 36 out of 47 counties in Kenya, including Murang'a, our study site [26]. Similarly, the Uganda Institute of Public Health reported confirmed RVF cases across 21 of 135 districts in Uganda from 2017 to 2023 [27]. These reports may have contributed to increased awareness and higher levels of knowledge about RVF in both countries. Although Uganda had the highest seroprevalence (10.4%), it did not have the highest knowledge levels. Transmission of RVF has been predominantly cryptic in the recent past and lack of awareness of ongoing RVF transmission has resulted in limited public education programs on the virus. Kenya

demonstrated high odds of knowledge and a much lower seroprevalence (2.0%), even though small RVF outbreaks are a common occurrence in the country, raising the possibility that community awareness on how to prevent RVF could have led to effective behavior modifications and lower rates of exposure. Recent analysis by our research group [14] detail the distribution and frequency of the RVF cases in Kenya's highlands, illustrating the virus's cryptic presence in non-epidemic settings. DRC, with the lowest seroprevalence (1.5%), also had the lowest odds of RVF knowledge. No RVF outbreak has been detected in DRC which then limits the use of communication and awareness campaigns regarding the disease. In addition, the risk of transmission of RVF virus has been much lower in Goma, DRC where we carried out our study, limiting the need for behavior modifications there to prevent transmission. Our findings suggest that several factors could influence awareness regarding the disease. These include exposure to the virus, the magnitude and frequency of outbreaks, the extent of public health campaigns, environmental factors, occupation and different practices. Future research should aim to identify and quantify these factors to inform tailored, country-specific strategies for RVF education and prevention.

We observed a positive association between mosquito bite prevention and RVF knowledge, aligning with previous studies. For instance [28] reported a positive correlation between RVF knowledge and the use of mosquito nets in Senegal yet [29] in Tanzania found that individuals with higher RVF knowledge were not necessarily more likely to use mosquito nets. Across the three countries, use of mosquito bite prevention methods was higher than RVF knowledge. This proposes that mosquito bite prevention may be better framed as a general health intervention that prevents the transmission of several mosquito borne diseases. Broad community-wide health interventions often have a greater impact on health improvement than disease-specific strategies alone [30].

We found that presence of livestock within homes was significantly associated with improved RVF knowledge, however, there was no association between individuals who reported animal farming to be their primary occupation and RVF knowledge. These findings suggest gaps in occupational health knowledge of RVF among animal farmers that needs to be addressed.

Our study had a few limitations. We relied on self-reported data which can lead to participants reporting socially favourable behaviours. One of the key strengths however is that the study was able to achieve and surpass the target sample size indicating better precision.

## Conclusions

In conclusion, our study highlights important cross-country differences in public awareness of RVF in East and Central Africa despite varying levels of disease exposure. Among the three countries studied, Kenya demonstrated the highest awareness likely due to more frequent outbreaks. Uganda showed moderate knowledge despite ongoing transmission, suggesting a disconnect between exposure risk and public health messaging. In contrast DRC exhibited the lowest awareness pointing to a critical gap in health education and outreach in regions with emerging or under-recognized transmission. Across all countries, key factors associated with advanced knowledge included older age, higher education, male sex and healthcare-related occupations. These findings highlight the need for targeted, country-specific, occupation-focused education strategies, particularly in areas where risk is high, but awareness remains low. Strengthening RVF communication and community engagement tailored to the local context will be key to reducing vulnerability and improving outbreak preparedness.

## Supporting information

**Supplementary Table 1. Description of the criteria of knowledge scoring.**
(DOCX)

**Supplementary Table 2. Bivariate analysis of factors associated with RVF knowledge levels among healthcare facility participants from Kenya, Uganda and DRC, 2022–2024.**
(DOCX)

**S1 Data. knowledge_dataset.**
(XLSX)

**S1 File. Inclusivity_questionnaire_20250602.**
(DOCX)

## Acknowledgments

We extend our heartfelt gratitude to all individuals and institutions that contributed to the success of this study. Special thanks go to the study participants for their invaluable cooperation and to our partners, including the Ministries of Health in Kenya, Uganda, and DRC. We also sincerely appreciate the guidance and support of our colleagues and collaborators throughout this research.

## Author contributions

**Conceptualization:** Raymond Odinoh, Jeanette Dawa, Silvia Situma, Luke Nyakarahuka, Luciana Lepore, Veerle Vanlerberghe, Carolyne Nasimiyu, Sheila Makiala, Daniel Mukadi, Barnabas Bakamutumaho, Justin Masumu, Robert F. Breiman, Kariuki Njenga.

**Data curation:** Raymond Odinoh, Christian Ifufa, Nicholas Owor.

**Formal analysis:** Raymond Odinoh, Jeanette Dawa.

**Funding acquisition:** Jeanette Dawa, Robert F. Breiman, Kariuki Njenga.

**Investigation:** Jeanette Dawa, Luke Nyakarahuka, Luciana Lepore, Deo Ndumu, Robert F. Breiman.

**Methodology:** Raymond Odinoh, Jeanette Dawa, Silvia Situma, Luke Nyakarahuka, Luciana Lepore, Veerle Vanlerberghe, Sheila Makiala, Barnabas Bakamutumaho, Justin Masumu, Robert F. Breiman, Kariuki Njenga.

**Project administration:** Raymond Odinoh, Jeanette Dawa, Silvia Situma, Luke Nyakarahuka, Luciana Lepore, Carolyne Nasimiyu, Christian Ifufa, Herve Viala, Nicholas Owor, Barnabas Bakamutumaho, Deo Ndumu, Justin Masumu.

**Resources:** Jeanette Dawa, Kariuki Njenga.

**Supervision:** Raymond Odinoh, Jeanette Dawa, Silvia Situma, Luke Nyakarahuka, Luciana Lepore, Carolyne Nasimiyu, Nicholas Owor, Justin Masumu.

**Validation:** Raymond Odinoh, Silvia Situma.

**Visualization:** Raymond Odinoh.

**Writing – original draft:** Raymond Odinoh, Jeanette Dawa, Kariuki Njenga.

**Writing – review & editing:** Raymond Odinoh, Jeanette Dawa, Silvia Situma, Luke Nyakarahuka, Luciana Lepore, Veerle Vanlerberghe, Carolyne Nasimiyu, Sheila Makiala, Christian Ifufa, Daniel Mukadi, Herve Viala, Barnabas Bakamutumaho, Justin Masumu, Robert F. Breiman, Kariuki Njenga.

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
