## [Decision Letter · Decision Letter 0]

26 Apr 2025

PONE-D-24-55546Have you heard of Rift Valley fever? Findings from a multi-country study in East and Central AfricaPLOS ONE?

Dear Dr. Odinoh,

Thank you for submitting your manuscript to PLOS ONE. After careful consideration, we feel that it has merit but does not fully meet PLOS ONE’s publication criteria as it currently stands. Therefore, we invite you to submit a revised version of the manuscript that addresses the points raised during the review process.

We look forward to receiving your revised manuscript.

Kind regards,

Timothy Omara

Academic Editor

PLOS ONE

Journal Requirements:

“We acknowledge the financial support provided by the U.S. National Institute of Allergy and Infectious Diseases/National Institutes of Health (NIAID/NIH), under grant number U01AI151799, through the Centre for Research in Emerging Infectious Diseases-East and Central Africa (CREID-ECA)”

“We acknowledge the financial support provided by the U.S. National Institute of Allergy and Infectious Diseases/National Institutes of Health (NIAID/NIH), under grant number U01AI151799, through the Centre for Research in Emerging Infectious Diseases-East and Central Africa (CREID-ECA).”

“We acknowledge the financial support provided by the U.S. National Institute of Allergy and Infectious Diseases/National Institutes of Health (NIAID/NIH), under grant number U01AI151799, through the Centre for Research in Emerging Infectious Diseases-East and Central Africa (CREID-ECA)”

6. We note that Figure 1 in your submission contain [map/satellite] images which may be copyrighted. All PLOS content is published under the Creative Commons Attribution License (CC BY 4.0), which means that the manuscript, images, and Supporting Information files will be freely available online, and any third party is permitted to access, download, copy, distribute, and use these materials in any way, even commercially, with proper attribution. For these reasons, we cannot publish previously copyrighted maps or satellite images created using proprietary data, such as Google software (Google Maps, Street View, and Earth). For more information, see our copyright guidelines: http://journals.plos.org/plosone/s/licenses-and-copyright.

7. Please ensure that you refer to Figure 1 in your text as, if accepted, production will need this reference to link the reader to the figure

Additional Editor Comments:

Dear Authors,

I have attached some corrections in the MS word file for your consideration. 

Reviewers' comments:

Reviewer's Responses to Questions

**Comments to the Author**

1. Is the manuscript technically sound, and do the data support the conclusions?

Reviewer #1: Yes

Reviewer #2: Partly

2. Has the statistical analysis been performed appropriately and rigorously?

Reviewer #1: Yes

Reviewer #2: Yes

3. Have the authors made all data underlying the findings in their manuscript fully available?

Reviewer #1: Yes

Reviewer #2: Yes

4. Is the manuscript presented in an intelligible fashion and written in standard English?

Reviewer #1: Yes

Reviewer #2: Yes

Reviewer #1: The authors can improve their manuscript by the following;

Abstract: In the conclusion of the abstract the authors should consider rewriting looking at their findings comparing the 3 different countries they looked at.

In the introduction, the authors mention infection through mosquito bites (Line 64, page 3), they can improve this by been exact on the type of mosquito that does the transmission of the infection.

Material and Methods: the authors should be exact on the types of kits used the detection of RVF in the study(line 144-145, page 7).

Conclusion: the authors should rewrite the conclusion, as it appears more as a discussion of the work now, by been more exact on their findings.

Reviewer #2: I have attached the correction on the assertion that sporadic detections have replaced the usual outbreaks related to weather. This wrong statement is found in the abstract. There is no evidence to support that assertion. The paper is good as it has included data across a region and national boundaries, as we know that RVF is a trans-boundary animal disease. The region has a diverse and heterogenous population and therefore the data is very relevant to all stakeholders especially policy makers and researchers

**Do you want your identity to be public for this peer review?** For information about this choice, including consent withdrawal, please see our Privacy Policy

Reviewer #1: No

Reviewer #2: **Yes: ** George Chege Gitao

---

## [Author Response · Author response to Decision Letter 1]

11 Jun 2025

RESPONSE TO REVIEWER COMMENTS

Manuscript title: Have you heard of Rift Valley fever? Findings from a multi-country study in East and Central Africa

Reviewer 1

1. Comment 1 (Line 23): Do you mean all the authors? If only some of the authors, then such authors should be marked as per PLOS ONE’s manuscript preparation guidelines

Response: Thank you for this comment. We have now specified the levels of author contributions.

2. Comment 2 (Line 30): This statement does not read proper to me. You could revised it to ‘‘Rift Valley Fever (RVF) has caused several outbreaks across Africa………..’’ which is more straightforward

Response: Thank you for the suggestion. We agree with the reviewer that the sentence could be made more direct. We have revised it to: "Rift Valley Fever (RVF) has caused several outbreaks across Africa, impacting human health and animal trade." (Line 32-34)

3. Comment 3 (Line 37): questions regarding their knowledge on?

Response: Thank you for the comment. We have clarified the sentence to specify that participants were asked questions regarding their knowledge of RVF transmission, symptoms, prevention, and control. The revised sentence now reads: "Sociodemographic information was collected, and participants were asked questions regarding their knowledge of RVF transmission, symptoms, prevention, and control." (Lines 39-42)

4. Comment 4 (Line 50): You could organize the data this way, so that the adjusted odds ratio remains in parenthesis

Response: We have reorganized the data presentation as recommended by including the adjusted odds ratio within the same parenthesis as the confidence interval

5. Comment 5 (Line 59): Do not repeat words that are already part of the title as author-suggested indexing keywords. It reduces article visibility

Response: Thank you for the insight, we have reworked this to maximize the article’s visibility. We have now excluded words already found in the title.

Reviewer 2

1. Comment 1 Line 31: Recently, sporadic detections among humans and animals in East Africa have replaced large scale outbreaks

Correction “This statement is not correct. It is known that epizootics occur and they are related to rainfall whereas sporadic outbreaks occur in between and this sustains the virus. Please table documented and verified evidence, that one has replaced the other.”

Response: Thank you for the correction. We acknowledge that sporadic detections have not replaced large-scale RVF outbreaks but rather occur between major epizootics, which are often associated with periods of heavy rainfall. We have revised the sentence accordingly.

The revised sentence now reads: "Recent reports indicate sporadic detections of RVF virus among humans and animals in East Africa during inter-epidemic periods. (Lines 34-35)

Reviewers Comments from HTML

Reviewer 1:

1. The authors can improve their manuscript by the following;

Abstract: In the conclusion of the abstract the authors should consider rewriting looking at their findings comparing the 3 different countries they looked at.

Response: We have revised the Abstract conclusion to incorporate a comparison of findings across the three countries studied. (Lines 61-67)

2. In the introduction, the authors mention infection through mosquito bites (Line 64, page 3), they can improve this by been exact on the type of mosquito that does the transmission of the infection.

Response: We have revised the Introduction to specify that RVF virus transmission occurs via bites from infected Aedes and Culex mosquitoes. (Line 76)

3. Material and Methods: the authors should be exact on the types of kits used the detection of RVF in the study (line 144-145, page 7).

Response: Thank you for the comment. While our original text referenced a companion paper describing the full testing protocol, we have now added the specific test kit name to the Materials and Methods section for easier reference. Full sensitivity, specificity, and quality control procedures remain available in the cited reference [19] (Line 158-164)

4. Conclusion: the authors should rewrite the conclusion, as it appears more as a discussion of the work now, by being more exact on their findings.

Response: We have revised the Conclusion section to focus more directly on the main findings, minimizing discussion points. (Lines 398-413)

Editor’s Comments

Reponse: We have taken this into consideration.

Response: This has been completed and included in the manuscript after the references but also attached separately

“We acknowledge the financial support provided by the U.S. National Institute of Allergy and Infectious Diseases/National Institutes of Health (NIAID/NIH), under grant number U01AI151799, through the Centre for Research in Emerging Infectious Diseases-East and Central Africa (CREID-ECA)”

Response:

Amended Funding Statement:

We acknowledge the financial support provided by the U.S. National Institute of Allergy and Infectious Diseases/National Institutes of Health (NIAID/NIH), under grant number U01AI151799, through the Centre for Research in Emerging Infectious Diseases-East and Central Africa (CREID-ECA). There was no additional external funding received for this study. The funders had no role in study design, data collection and analysis, decision to publish, or preparation of the manuscript.

Please confirm at this time whether or not your submission contains all raw data required to replicate the results of your study. Authors must share the “minimal data set” for their submission. PLOS defines the minimal data set to consist of the data required to replicate all study findings reported in the article, as well as related metadata and methods (https://journals.plos.org/plosone/s/data-availability#loc-minimal-data-set-definition). For example, authors should submit the following data:

Response: All relevant data underlying the findings of this study are provided in the Supporting Information files. These include de-identified survey responses, laboratory test results, and statistical outputs necessary to replicate the reported analyses. Due to ethical restrictions regarding participant confidentiality, individual-level survey data have been de-identified. Further inquiries may be directed to the PI. After publication, the data will be further uploaded to Havard repository under WSU-GHK account. https://dataverse.harvard.edu/dataverse/wsughk

5. Thank you for stating the following in the Acknowledgments Section of your manuscript: “We acknowledge the financial support provided by the U.S. National Institute of Allergy and Infectious Diseases/National Institutes of Health (NIAID/NIH), under grant number U01AI151799, through the Centre for Research in Emerging Infectious Diseases-East and Central Africa (CREID-ECA).”

“We acknowledge the financial support provided by the U.S. National Institute of Allergy and Infectious Diseases/National Institutes of Health (NIAID/NIH), under grant number U01AI151799, through the Centre for Research in Emerging Infectious Diseases-East and Central Africa (CREID-ECA)”

Response: Moved it to its own section as funding statement and also included the statement within the cover letter

6. We note that Figure 1 in your submission contain [map/satellite] images which may be copyrighted. All PLOS content is published under the Creative Commons Attribution License (CC BY 4.0), which means that the manuscript, images, and Supporting Information files will be freely available online, and any third party is permitted to access, download, copy, distribute, and use these materials in any way, even commercially, with proper attribution. For these reasons, we cannot publish previously copyrighted maps or satellite images created using proprietary data, such as Google software (Google Maps, Street View, and Earth). For more information, see our copyright guidelines: http://journals.plos.org/plosone/s/licenses-and-copyright.We require you to either (1) present written permission from the copyright holder to publish these figures specifically under the CC BY 4.0 license, or (2) remove the figures from your submission:

We recommend that you contact the original copyright holder with the Content Permission Form (http://journals.plos.org/plosone/s/file?id=7c09/content-permission-form.pdf) and the following text: “I request permission for the open-access journal PLOS ONE to publish XXX under the Creative Commons Attribution License (CCAL) CC BY 4.0 (http://creativecommons.org/licenses/by/4.0/). Please be aware that this license allows unrestricted use and distribution, even commercially, by third parties. Please reply and provide explicit written permission to publish XXX under a CC BY license and complete the attached form.”

Response: Thank you for your attention to copyright compliance. We confirm that Figure 1 was self-generated using QGIS software, based on publicly available shapefiles from Natural Earth and GADM databases. No copyrighted basemaps such as Google Maps, Google Earth, or ESRI layers were used. The figure is therefore fully compliant with the CC BY 4.0 license.

7. Please ensure that you refer to Figure 1 in your text as, if accepted, production will need this reference to link the reader to the figure

Response: We have added the statement “The geographic distribution of the study areas is shown in Figure 1” under study areas to reference the figure.

Response: The only change in the reference list is change of reference 19 which was initially a preprint but now a peer reviewed publication.

---

## [Editor Report · Decision Letter 1]

16 June 2025

Have you heard of Rift Valley fever? Findings from a multi-country study in East and Central Africa

PONE-D-24-55546R1

Dear Dr. Odinoh,

We’re pleased to inform you that your manuscript has been judged scientifically suitable for publication and will be formally accepted for publication once it meets all outstanding technical requirements.

Kind regards,

Timothy Omara

Academic Editor

PLOS ONE
---

## [Editor Report · Acceptance letter]

PONE-D-24-55546R1

PLOS ONE

Dear Dr. Odinoh,

I'm pleased to inform you that your manuscript has been deemed suitable for publication in PLOS ONE. Congratulations! Your manuscript is now being handed over to our production team.

Kind regards,

on behalf of

Dr. Timothy Omara

Academic Editor

PLOS ONE